# Adherence to Opioid Patient Prescriber Agreements at a Safety Net Hospital

**DOI:** 10.3390/cancers15112943

**Published:** 2023-05-27

**Authors:** Soraira Pacheco, Linh M. T. Nguyen, John M. Halphen, Nikitha N. Samy, Nathaniel R. Wilson, Gregory Sattler, Shane E. Wing, Christine Feng, Rex A. D. Paulino, Pulin Shah, Supriyanka Addimulam, Riddhi Patel, Curtis J. Wray, Joseph A. Arthur, David Hui

**Affiliations:** 1Joan and Stanford Alexander Division of Geriatric and Palliative Medicine, McGovern Medical School, UTHealth, Houston, TX 77030, USA; 2McGovern Medical School, The University of Texas Health Science Center at Houston, Houston, TX 77030, USA; 3Department of Epidemiology, Human Genetics & Environmental Sciences, The University of Texas Health Science Center at Houston School of Public Health, Houston, TX 77030, USA; 4Department of Palliative Care, Rehabilitation and Integrative Medicine, The University of Texas MD Anderson Cancer Center, Houston, TX 77030, USA

**Keywords:** analgesics, opioid, cancer pain, neoplasm, opioid-related disorders, illicit drugs, pain management, palliative care, contract

## Abstract

**Simple Summary:**

Opioids are often used to treat cancer-related pain. Non-medical opioid use (NMOU) is a potential concern in all patients. Patient prescriber agreements (PPAs) have been recommended as a risk mitigation strategy. However, few studies have examined their use. The aim of our retrospective study was to assess how often PPAs were completed and how often cancer patients did not adhere to the PPA in a palliative care clinic. We found that 54% of patients had a PPA, and 10% were not adherent. PPAs were associated with younger age and alcohol use. Non-adherence was associated with males, being single, tobacco and alcohol use, contact with persons involved in criminal activity, opioid use for non-cancer pain, and higher pain score. PPA non-adherence occurred in a minority of patients, particularly those with NMOU risk factors. Our findings support the potential role of universal PPAs and systematic screening of NMOU risk factors.

**Abstract:**

Patient prescriber agreements, also known as opioid contracts or opioid treatment agreements, have been recommended as a strategy for mitigating non-medical opioid use (NMOU). The purpose of our study was to characterize the proportion of patients with PPAs, the rate of non-adherence, and clinical predictors for PPA completion and non-adherence. This retrospective study covered consecutive cancer patients seen at a palliative care clinic at a safety net hospital between 1 September 2015 and 31 December 2019. We included patients 18 years or older with cancer diagnoses who received opioids. We collected patient characteristics at consultation and information regarding PPA. The primary purpose was to determine the frequency and predictors of patients with a PPA and non-adherence to PPAs. Descriptive statistics and multivariable logistic regression models were used for the analysis. The survey covered 905 patients having a mean age of 55 (range 18–93), of whom 474 (52%) were female, 423 (47%) were Hispanic, 603 (67%) were single, and 814 (90%) had advanced cancer. Of patients surveyed, 484 (54%) had a PPA, and 50 (10%) of these did not adhere to their PPA. In multivariable analysis, PPAs were associated with younger age (odds ratio [OR] 1.44; *p* = 0.02) and alcohol use (OR 1.72; *p* = 0.01). Non-adherence was associated with males (OR 3.66; *p* = 0.007), being single (OR 12.23; *p* = 0.003), tobacco (OR 3.34; *p* = 0.03) and alcohol use (OR 0.29; *p* = 0.02), contact with persons involved in criminal activity (OR 9.87; *p* < 0.001), use for non-malignant pain (OR 7.45; *p* = 0.006), and higher pain score (OR 1.2; *p* = 0.01). In summary, we found that PPA non-adherence occurred in a substantial minority of patients and was more likely in patients with known NMOU risk factors. These findings underscore the potential role of universal PPAs and systematic screening of NMOU risk factors to streamline care.

## 1. Introduction

Pain is one of the most common symptoms in patients with cancer, occurring in 50% to 90% of patients with advanced disease [1,2]. For severe chronic cancer pain, opioids are often considered frontline therapy by the palliative care team. They can effectively relieve pain but also carry significant risks for non-medical opioid use (NMOU). NMOU is defined as use without a prescription or in greater amounts, more often, or longer than prescribed, or for a reason other than a doctor’s instructions [3]. NMOU, particularly opioid use disorder, is associated with significant distress and overdose deaths [4,5,6]. However, in patients with cancer, the best practices for identifying and optimally managing NMOU have not been clearly defined [7,8,9].

Patient prescriber agreements (PPAs), also known as opioid contracts or opioid treatment agreements, have been recommended as a risk mitigation strategy for NMOU in cancer and non-cancer patients [10,11,12]. Despite the limited evidence, several guidelines recommend the adoption of PPAs. The use of a written PPA with patients outlines the risks and benefits of treatment, procedures for safeguarding opioids in the home, and safe opioid use practices. Some documents specifically state that patients shall not share or sell their opioids. To date, PPAs have mainly been used in chronic non-malignant pain settings [13,14], with mixed evidence regarding their effectiveness. Only a handful of case studies have examined the use of PPAs in patients with cancer [15,16,17]. PPAs have not been well characterized, particularly in the palliative care setting [14,18,19]. To our knowledge, no studies have examined PPAs in cancer palliative care settings in safety net hospitals, where the patients may be at higher risk of NMOU. A better understanding would help improve opioid stewardship and the palliative care approach in patients with cancer.

Harris Health System’s Lyndon B. Johnson (LBJ) General Hospital and Outpatient Center is a county hospital and clinic serving ethnically diverse, underinsured, and uninsured patients. Many patients seen at LBJ hospital have mental illness and substance use disorders. In the palliative care clinic, we have implemented a safe opioid use program in which patients starting on opioids were asked to sign a PPA as a risk mitigation strategy. All faculty were encouraged to collect PPAs from patients on the initial clinic visit. Our study examined the proportion of patients who did not adhere to PPAs, reasons for non-adherence, and clinical predictors. We also examined the patient characteristics associated with establishing PPAs and non-adherence to PPAs.

## 2. Materials and Methods

### 2.1. Design

This is a retrospective study of consecutive patients seen by the LBJ palliative care team for consultation between 1 September 2015 and 31 December 2019. We included consecutive patients who were 18 years or older, had a cancer diagnosis, and received opioid prescriptions. The institutional review board approved the study protocol at McGovern Medical School and Harris Health System with a waiver of informed consent.

### 2.2. Data Collection

We collected patient demographic information such as medical record number, date of birth, sex, race, marital status, cancer diagnosis and stage, insurance, socioeconomic status information, and risk factors for NMOU such as a history of a substance use disorder, marijuana use, tobacco use, and alcoholism [20,21]. We also assessed psychiatric comorbidities and family history of substance use disorder. In addition, we retrieved data on symptom burden using the Edmonton Symptom Assessment Scale (ESAS). ESAS measures the average symptom intensity in the past 24 h using a 0–10 point numeric rating scale, where 0 = no symptoms and 10 = the worst possible symptoms (such as pain, fatigue, nausea, depression, anxiety, drowsiness, shortness of breath, loss of appetite, poor sleep, loss of feeling of well-being, constipation, family distress, financial distress, and spiritual pain) [21,22]. 

The PPA is a one-page document that describes the patient’s role and responsibilities (Appendix A). It states that the patient agrees to have only one prescriber and one pharmacy. PPA also states that the patient “will not share or sell narcotic pain killers with others nor will they not obtain controlled substances from any other people or doctors”. The PPA states that urine drug screens (UDS) might be requested “from time to time [and] If I have illicit drugs or drugs in my system that were not prescribed for me, my doctor may refer me for treatment for addiction, discontinue opioid treatment of my chronic pain, and/or discharge me from their practice”. This form was available in both English and Spanish. 

For each patient, we conducted a chart review to determine if a PPA was completed, the date of completion if applicable, if the patient was adherent, and if not, the date of non-adherence (i.e., an indicator of NMOU). We also collected the reasons for PPA non-adherence. NMOU behaviors were based on a chart review of all patient consultations and follow-up visits with our palliative care clinic. Reasons for PPA non-adherence were classified under three major categories: (1) opioid use pattern inconsistent with palliative care team recommendations (e.g., excessive self-escalation of opioids, using pain medications for reasons other than pain, asking for an early opioid refill, resistance to change in opioid regimen), (2) UDS abnormalities (e.g., positive screens for amphetamine, barbiturates, benzodiazepines, cannabinoids, cocaine, or PCP, or negative for prescribed opioids) [23], and (3) other (e.g., doctor shopping or unauthorized multiple prescribers, stolen or lost prescription, obtaining opioid from a non-medical source or stealing).

### 2.3. Statistical Analysis

We summarized patient characteristics using descriptive measures such as counts, percentages, means, standard deviations, median and interquartile ranges. We then examined the association between patient characteristics and PPA completion status (yes or no) with univariate logistic regression models. We also conducted multivariable logistic regression analysis with purposeful variable selection to include relevant clinical variables that were statistically significant (i.e., age, sex, race/ethnicity, cancer diagnosis), history of illicit drug use, history of marijuana use, history of tobacco use, history of alcohol use, personal history of criminal activity, and contact with persons involved in criminal activity).

We conducted a similar univariable and multivariable logistic regression analysis for the outcome of non-adherence among patients who completed PPA. The variables included in the multivariable model were male gender, black non-Hispanic, history of illicit drug use, history of marijuana use, history of tobacco use, history of alcohol use, history of schizophrenia, personal history of criminal activity, contact with persons involved in criminal activity, inconsistent pain presentation, use for non-malignant pain, and pain score at consult. Additionally, the frequency and percentage of reasons for non-adherence to PPA were also evaluated. Statistical analyses were conducted using STATA SE version 17.0 (College Station, TX, USA). A *p*-value of < 0.05 was considered to be statistically significant.

## 3. Results

### 3.1. Patient Characteristics

A total of 905 consecutive patients were included. The mean age was 55 (range 18–93). Of the patients, 590 (65%) were under the age of sixty, 474 (52%) were female, 423 (47%) were of Hispanic ethnicity, 603 (67%) were single, and 814 (90%) had advanced cancer. The median interquartile range (IQR) follow-up time was 57 days (Q1–Q3 1–213 days). The risk factors of NMOU at baseline were shown in Table 1, with a history of illicit drug use (*n* = 160, 17.7%), tobacco use (*n* = 453, 50.1%), marijuana use (*n* = 197, 21.8%), and alcohol use (*n* = 172, 19%) being the most common. The average pain rating on ESAS at consultation was 5.8 (SD = 3.4).

### 3.2. Completion of PPA

The palliative care physicians discussed PPAs with 484 patients, and 100% of the patients signed the document. Of all patients, 484 (54%) had a completed PPA documented in the electronic health record (Table 2). The median time between palliative care consultation and PPA completion was 0 days (*n* = 484, Q1–Q3 0–20 days). 

In univariate analysis, completion of PPA was associated with younger age, male gender, White non-Hispanic, Black non-Hispanic, head and neck cancer, history of illicit drug use, history of marijuana use, history of tobacco use, history of alcohol use, and personal history of criminal activity and pain (Table 2).

In multivariable analysis, patients younger than 60 years of age had 1.44 (95% CI: 1.07–1.4) times the odds of having completed the PPA as compared to patients 60 or more years old (Table 2). Moreover, having a history of alcohol use was associated with the completion of the PPA (OR: 1.72, 95% CI: 1.12–2.64) (Table 2).

### 3.3. Non-Adherence to PPA among Patients Who Completed PPA

Fifty patients (10%) did not adhere to the PPA (Table 1). The median (IQR) time between PPA completion and non-adherence was 57 days (*n* = 50, Q1–Q3 7–161 days) among all patients with a PPA. Of these, 38 (76%) of patients were non-adherent to the PPA on a different day than the day of the PPA, and the median (IQR) time between PPA completion and non-adherence was 70 days (*n* = 38, Q1–Q3 35–207 days) among those patients. 

In univariate analysis, non-adherence to the PPA was associated with male gender, Black non-Hispanic identity, being single, history of illicit drug use, history of marijuana use, history of tobacco use, history of schizophrenia, personal history of criminal activity, contact with persons involved in criminal activity, inconsistent pain presentation, use for non-malignant pain, other risk factors of NMOU, and higher ESAS scores at consultation (pain, depression, anxiety, appetite, and family distress) (Table 3). 

In multivariable analysis, non-adherence to the PPA was associated with male gender (OR: 3.66, 95% CI: 1.43–9.32), being single (OR: 12.23, 95% CI: 2.29–65.45), history of tobacco use (OR: 3.34, 95% CI: 1.12–9.99), history of alcohol use (OR: 0.29, 95% CI: 0.1–0.79), contact with persons involved in criminal activity (OR: 9.86, 95% CI: 2.75–35.38), use for non-malignant pain medication (OR 7.45, 95% CI: 1.8–30.9) and higher ESAS pain expression at consultation (OR: 1.2, 95% CI: 1.04–1.38) (Table 3). 

The main reasons for non-adherence were not following prescription instructions (*n* = 20, 40%), urine drug screen abnormalities (*n* = 18, 36%), and other (e.g., doctor shopping, lost prescription, obtaining opioids from non-medical source) (*n* = 22, 44%).

## 4. Discussion

In this study of cancer patients seen at a palliative care clinic at a safety net hospital, approximately half had an established PPA. Our data showed that patients who were younger and had a history of alcohol use were more likely to have signed a PPA. Approximately one in ten patients did not adhere to the PPA. Predictors of non-adherence included male gender, being single, tobacco and alcohol use, contact with persons involved in criminal activity, and use for non-malignant pain. Based on these findings, we propose PPAs with risk stratification to monitor patients at high risk of non-adherence. 

There is much debate about whether PPAs are helpful as a risk mitigation strategy for NMOU. There have been only two systematic reviews in the past two decades, both showing weak evidence supporting the effectiveness of PPAs in reducing and mitigating opioid misuse and abuse [10,11,12,13,14,24,25]. Proponents of PPAs suggest that they support patient autonomy and shared decision-making by allowing the patient to assess the benefits, risks, and limitations of opioid medications. Critics of PPAs argue that PPAs can undermine autonomy and impair trust in physician-patient relationships, as the language in PPAs may be perceived as mistrustful and accusatory, which may stigmatize the patient and erode trust in the patient-physician relationship [14,24,26,27,28]. The evidence for PPAs being beneficial as an opioid management strategy in cancer patients is even more limited, with only a few case reports available [8,9,10,11,17,24,25]. 

Despite our departmental efforts to promote the routine use of PPAs, only half of the patients had a signed PPA. There are multiple barriers to systematically administering PPAs, which can be classified under clinician, patient, and system factors. Clinician factors included lack of time, interest, and confidence in the effectiveness of PPAs. Patient factors included lack of time to sign the PPA, language barriers, and ambivalence about signing a contract. System factors included the lack of prompting, lack of hospital standards/policy, lack of processes for administering the forms, and lack of integration of PPA in the electronic medical record. If PPAs were implemented universally, it could reduce stigma and minimize variations in practice [14,26,28]. 

Our data showed that PPAs were conducted in a more targeted fashion in our clinic, focusing on younger individuals with a history of alcohol use. These factors suggest clinicians were likely identifying patients at risk of NMOU. Interestingly, non-Hispanic Blacks, non-Hispanic Whites, males, and patients with head and neck cancer were more likely to have signed a PPA in univariable analysis, suggesting that there may be implicit bias [28,29]. Language barriers and time for interpretation may also contribute to the discrepancy. Universal PPAs may minimize these disparities.

To our knowledge, this is the largest series to date to examine non-adherence to PPAs in the cancer palliative care setting. Of those who signed a PPA, 10% were documented to be non-adherent throughout the monitoring period. In most studies evaluating non-adherence to PPAs, almost one-half of non-adherence was due to inconsistent use of controlled substances [30]. In our clinic, reasons for non-adherence to PPA were not following instructions according to the prescription, urine drug screen abnormalities, and other. The relatively low non-adherence rate to PPAs may be an underrepresentation, since some NMOU behaviors are not easily detected. Even when detected, the clinicians may consider other factors before deciding to discontinue opioids or dismiss the patient from the clinic. We operate within a public healthcare safety net system, and if a patient does not adhere to the PPA, the palliative care team does not automatically stop prescribing opioids or terminate the patient–clinician relationship. Instead, our team discusses the patient’s case with a Pain Board before making a formal decision to document non-adherence. The palliative care team provides options such as tapering opioids, and we continue to see the patient in the clinic to manage other symptoms and provide non-opioid pharmacological interventions. However, 10% is a significant minority and highlights the need to closely monitor high-risk patients in the cancer palliative care setting [9]. 

There is a significant overlap between NMOU and PPA non-adherence. NMOU includes a range of behaviors, such as aberrant UDS, excessive opioid use, substance use disorder, and diversion [31,32]. PPA non-adherence, by definition, is a form of NMOU. However, not all NMOU may rise to the threshold of PPA non-adherence. For example, a patient who misunderstood prescription instructions and took the opioids not as prescribed could be considered NMOU but still adherent to the PPA by an understanding clinician. 

A recent study of cancer patients found that being single, having a MEDD greater than 50 mg, and having SOAPP scores greater than 7 were associated with a higher risk for the presence of NMOU behavior [29,33]. In our clinic, factors associated with non-adherence to PPAs were male gender, being single, smoking history, contact with persons involved in criminal activity, and use for non-malignant pain. Our findings are generally consistent with the literature. In cancer patients, factors associated with PPA non-adherence were younger age, personal or familial mental health history, and history of illicit drug use [29,33,34,35]. In patients with non-cancer chronic pain, factors associated with PPA non-adherence include active tobacco use, prior driving while intoxicated, drug-related offenses, being younger, and having an underlying psychiatric disorder [24,26,36,37]. This highlights the importance of identifying risk factors at baseline and providing risk stratification for monitoring. As clinical practice guidelines recommend, patients with multiple risk factors may need more frequent UDS and clinic visits to support safe opioid use [21]. 

## 5. Conclusions

Our results underscore that PPAs can be easily administered with a large sample size even in resource-limited settings. A substantial minority of patients studied had NMOU behaviors and non-adherence to PPAs, and we identified key patient risk factors associated with PPA non-adherence. 

Given our high-risk population, we believe universal PPAs should be promoted as one risk mitigation strategy to promote safe opioid use. Although the evidence that PPAs reduce adverse opioid-related events is limited, PPAs can be helpful as educational material, setting patient expectations for testing requirements and providing reassurance on the partnership, especially when many patients have an opioid phobia. PPAs can be used as a form of monitoring and documenting NMOU behavior. Furthermore, implementing a more universal protocol could help reduce clinicians’ implicit bias in providing opioid pain management. 

Our study has several strengths, including consecutive patient data collection, a large, ethnically diverse sample size, and a unique focus on a cancer palliative care setting in a safety net hospital. However, our study had several limitations. There was a lack of generalizability, as the data was obtained from a single palliative care clinic in a single hospital system. Additionally, we could not determine if PPA reduces NMOU behavior or overdoses. The retrospective study design did not allow us to capture important predictors such as patient attitudes and beliefs regarding PPAs. Among the patients who did not complete PPAs, we could not tell from the chart whether it was because PPAs were not offered, or the patient refused to sign. Thus, we could not evaluate the effectiveness of PPAs in this study. A prospective study may improve data collection, but having to obtain patient consent may result in selection bias. Future research, such as randomized controlled trials, is needed to assess the benefit of universal PPAs in reducing NMOU. In this era of the opioid epidemic, the management of patients on opioids is highly complex, and PPAs may potentially represent a low-cost and simple intervention among other risk mitigation strategies to support safe opioid use.

## Figures and Tables

**Table 1 cancers-15-02943-t001:** Patient Demographics by Completion of PPA and Non-Adherence to PPA.

		PPA Completed		PPA Non-Adherent	
Demographic Characteristics	Total *n* = 905	Yes *n* = 484 (53.5%)	No *n* = 421 (46.5%)	Yes *n* = 50 (10.3%)	No *n* = 434 (89.7%)
Age (in years)					
Mean (range)	55.1 (18–93)	54.0 (20–89)	56.3 (18–93)	53.1 (28–68)	54.1 (20–89)
Age (in years)					
<60	590 (65.2%)	337 (69.6%)	253 (60.1%)	34 (68.0%)	303 (69.8%)
≥60	315 (34.8%)	147 (30.4%)	168 (39.9%)	16 (32.0%)	131 (30.2%)
Sex, n (%)					
Female	474 (52.4%)	237 (49.0%)	237 (56.3%)	15 (30.0%)	222 (51.2%)
Male	431 (47.6%)	247 (51.0%)	184 (43.7%)	35 (70.0%)	212 (48.9%)
Race/ethnicity, *n* (%)					
Hispanic, any race	423 (46.7%)	204 (42.2%)	219 (52.0%)	10 (20.0%)	194 (44.7%)
White, non-Hispanic	202 (22.3%)	118 (24.4%)	84 (20.0%)	11 (22.0%)	107 (24.7%)
Black, non-Hispanic	257 (28.4%)	152 (31.40%)	105 (25.0%)	28 (56.0%)	124 (28.6%)
Other race, non-Hispanic	23 (2.5%)	10 (2.1%)	13 (3.1%)	1 (2.0%)	9 (2.1%)
Marital status, *n* (%)					
Single *	603 (66.6%)	330 (68.2%)	273 (64.9%)	46 (92.0%)	284 (65.4%)
Married	302 (33.4%)	154 (31.8%)	148 (35.2%)	4 (8.0%)	150 (34.6%)
Cancer type, *n* (%)					
Gastrointestinal	312 (34.5%)	158 (32.6%)	154 (36.6%)	17 (34.0%)	141 (32.5%)
Respiratory	108 (11.9%)	64 (13.2%)	44 (10.5%)	4 (8.0%)	60 (13.8%)
Gynecological	103 (11.4%)	50 (10.3%)	53 (12.6%)	1 (2.0%)	49 (11.3%)
Genitourinary	96 (10.6%)	58 (12.0%)	38 (9.0%)	8 (16.0%)	50 (11.5%)
Breast	92 (10.1%)	46 (9.5%)	46 (10.9%)	5 (10.0%)	41 (9.5%)
Head and neck	81 (9.0%)	52 (10.0%)	29 (6.9%)	8 (16.0%)	44 (10.1%)
Hematological	57 (6.3%)	30 (6.2%)	27 (6.4%)	2 (4.0%)	28 (6.4%)
Other	56 (6.2%)	26 (5.4%)	30 (7.1%)	5 (10.0%)	21 (4.8%)
Cancer stage, *n* (%)					
Metastatic	607 (67.1%)	326 (67.4%)	281 (66.8%)	29 (58.0%)	297 (68.4%)
Locally advanced	187 (20.7%)	102 (21.1%)	85 (20.2%)	16 (32.0%)	86 (19.8%)
Localized	90 (9.9%)	44 (9.1%)	46 (10.9%)	5 (10.0%)	39 (9.0%)
Recurrent	15 (1.7%)	10 (2.1%)	5 (1.2%)	0 (0.0%)	10 (2.3%)
Advanced	5 (0.5%)	2 (0.4%)	3 (0.7%)	0 (0.0%)	2 (0.5%)
First line	1 (0.1%)	0 (0.0%)	1 (0.2%)	-	-
Risk factors of NMOU, n (%)					
History of illicit drug use	160 (17.7%)	107 (22.1%)	53 (12.6%)	29 (58.0%)	78 (18.0%)
History of marijuana use	197 (21.8%)	135 (27.9%)	62 (14.8%)	26 (52.0%)	109 (25.1%)
History of tobacco use	453 (50.1%)	273 (56.4%)	180 (42.9%)	43 (86.0%)	230 (53.0%)
History of alcohol use	172 (19.0%)	119 (24.6%)	53 (12.6%)	18 (36.0%)	101 (23.3%)
History of depression	155 (17.2%)	91 (18.8%)	64 (15.2%)	12 (24.0%)	79 (18.2%)
History of bipolar disorder	23 (2.5%)	12 (2.5%)	11 (2.6%)	3 (6.0%)	9 (2.1%)
History of schizophrenia	8 (0.9%)	6 (1.2%)	2 (0.5%)	3 (6.0%)	3 (0.7%)
Family history of illicit drug use	16 (1.8%)	10 (2.1%)	6 (1.4%)	3 (6.0%)	7 (1.6%)
Personal history of criminal activity	85 (9.4%)	58 (12.0%)	27 (6.4%)	22 (44.0%)	36 (8.3%)
Contact with persons involved in criminal activity	59 (6.5%)	40 (8.2%)	19 (4.5%)	20 (40.0%)	20 (4.6%)
Inconsistent pain presentation	15 (1.7%)	9 (1.9%)	6 (1.4%)	8 (16.0%)	1 (0.2%)
Use for non-malignant pain	36 (4.0%)	17 (3.5%)	19 (4.5%)	7 (14.0%)	10 (2.3%)
Others **	6 (0.7%)	4 (0.8%)	2 (0.5%)	3 (6.0%)	1 (0.2%)
MEDD at consult, median (Q1–Q3)	30 (10–70)	40 (14–75)	30 (5–62)	30 (10–60)	40 (15–80)
ESAS at consult, mean (SD)					
Pain	5.7 (3.4)	5.9 (3.4)	5.4 (3.5)	7.2 (2.7)	5.8 (3.4)
Tiredness	5.3 (3.4)	5.2 (3.3)	5.4 (3.4)	5.7 (3.0)	5.1 (3.3)
Nausea	2.4 (3.1)	2.4 (3.0)	2.5 (3.2)	3.0 (3.5)	2.3 (2.9)
Depression	2.7 (3.3)	2.5 (3.1)	2.9 (3.5)	3.5 (3.4)	2.4 (3.1)
Anxiety	2.9 (3.5)	2.8 (3.4)	3.0 (3.6)	4.0 (3.5)	2.7 (3.4)
Drowsiness	2.8 (3.3)	2.6 (3.1)	3.1 (3.5)	2.6 (2.9)	2.6 (3.2)
Appetite	3.9 (3.5)	4.0 (3.5)	3.8 (3.6)	5.2 (3.7)	3.9 (3.4)
Well-being	3.7 (3.4)	3.7 (3.4)	3.8 (3.5)	4.1 (3.5)	3.6 (3.3)
Shortness of breath	2.7 (3.3)	2.8 (3.2)	2.7 (3.3)	3.6 (3.3)	2.7 (3.2)
Family distress	2.0 (3.1)	2.0 (3.1)	2.1 (3.1)	3.0 (3.6)	1.9 (3.0)
Spiritual distress	1.3 (2.5)	1.2 (2.3)	1.5 (2.7)	1.5 (2.4)	1.2 (2.3)
Constipation	2.9 (3.5)	2.9 (3.5)	2.8 (3.5)	3.2 (3.7)	2.9 (3.4)
Sleep	4.8 (3.5)	4.9 (3.5)	4.7 (3.6)	6.0 (3.6)	4.9 (3.5)

* Marital status: single category includes unmarried, divorced, separated, and widowed. ** Others: homelessness, history of sexual abuse. Abbreviations: ESAS, Edmonton Symptom Assessment System; MEDD, morphine equivalent daily dose; NMOU, non-medical opioid use; PPA, Patient Prescriber Agreement.

**Table 2 cancers-15-02943-t002:** Predictors of Patients with a PPA.

	Univariate Analysis		Multivariable Analysis	
	Unadjusted Odds Ratio (95% Confidence Interval)	*p*-Value	Adjusted Odds Ratio (95% Confidence Interval)	*p*-Value
Age				
<60 years	1.52 (1.16–2.00)	0.003	1.44 (1.07–1.4)	0.02
≥60 years	Ref		-	
Sex				
Male	1.4 (1.03–1.75)	0.03	1.09 (0.80–1.49)	0.57
Female	Ref		Ref	
Race/ethnicity				
White, non-Hispanic	1.51 (1.08–2.12)	0.02	1.09 (0.74–1.62)	0.66
Black, non-Hispanic	1.56 (1.14–2.13)	0.006	1.30 (0.91–1.86)	0.15
Other race, non-Hispanic	0.83 (0.35–1.92)	0.66	0.75 (0.31–1.81)	0.52
Hispanic, any race	Ref		Ref	
Marital status				
Single	1.16 (0.88–1.53)	0.29	0.98 (0.72–1.34)	0.89
Married	Ref		Ref	
Cancer stage				
Locally advanced	1.03 (0.74–1.44)	0.84	-	-
Localized	0.82 (0.53–1.28)	0.39	-	-
Recurrent	1.71 (0.58–5.10)	0.33	-	-
Advanced	0.57 (0.95–3.46)	0.55	-	-
Metastatic	Ref	-	-	-
Risk factors of NMOU				
History of illicit drug use	1.97 (1.37–2.81)	<0.001	0.99 (0.58–1.67)	0.96
History of marijuana use	2.23 (1.60–3.12)	<0.001	1.33 (0.87–2.02)	0.18
History of tobacco use	1.73 (1.33–2.25)	<0.001	1.27 (0.92–1.77)	0.15
History of alcohol use	2.26 (1.58–3.22)	<0.001	1.72 (1.12–2.64)	0.01
History of depression	1.29 (0.91–1.83)	0.16	-	-
History of bipolar disorder	0.95 (0.41–2.17)	0.89	-	-
History of schizophrenia	2.62 (0.53–13.07)	0.24	1.28 (0.24–6.93)	0.77
Family history of illicit drug use	1.46 (0.52–4.04)	0.47	-	-
Personal history of criminal activity	1.98 (1.23–3.19)	0.005	1.24 (0.59–2.60)	0.56
Contact with persons involved in criminal activity	1.90 (1.08–3.34)	0.03	1.02 (0.47–2.24)	0.96
Inconsistent pain presentation	1.31 (0.46–3.70)	0.61	-	-
Use for non-malignant pain	0.77 (0.39–1.50)	0.44	0.57 (0.28–1.19)	0.13
Others **	1.74 (0.32–9.56)	0.52	-	-
MEDD at consult only	1.00 (1.00–1.00)	0.27	-	-
ESAS at consult only				
Pain	1.05 (1.01–1.09)	0.02	1.03 (0.98–1.07)	0.21
Tiredness	0.98 (0.94–1.02)	0.43	-	-
Nausea	0.99 (0.95–1.04)	0.70	-	-
Depression	0.97 (0.93–1.01)	0.09	-	-
Anxiety	0.99 (0.95–1.03)	0.47	-	-
Drowsiness	0.96 (0.92–1.00)	0.07	-	-
Appetite	1.01 (0.97–1.05)	0.53	-	-
Well-being	0.99 (0.95–1.03)	0.62	-	-
Shortness of breath	1.01 (0.97–1.05)	0.66	-	-
Family distress	0.99 (0.95–1.04)	0.67	-	-
Spiritual distress	0.95 (0.90–1.00)	0.07	-	-
Constipation	1.00 (0.97–1.05)	0.82	-	-
Sleep	1.01 (0.98–1.05)	0.47	-	-

** Others: homelessness, history of sexual abuse. Abbreviations: ESAS, Edmonton Symptom Assessment System; MEDD, morphine equivalent daily dose; NMOU, non-medical opioid use; PPA, Patient Prescriber Agreement.

**Table 3 cancers-15-02943-t003:** Association between non-adherence to a Patient Prescriber Agreement and patient characteristics.

	Univariate Analysis		Multivariable Analysis	
	Unadjusted Odds Ratio (95% Confidence Interval)	*p*-Value	Adjusted Odds Ratio (95% Confidence Interval)	*p*-Value
Age				
<60 years	0.92 (0.49–1.72)	0.79	0.93 (0.38–2.26)	0.87
≥60 years	Ref		-	
Sex				
Male	2.44 (1.30–4.60)	0.006	3.66 (1.43–9.32)	0.007
Female	Ref		Ref	
Race/ethnicity				
White, non-Hispanic	1.99 (0.82–4.85)	0.13	0.61 (0.19–1.96)	0.41
Black, non-Hispanic	4.38 (2.06–9.33)	<0.001	1.74 (0.65–4.72)	0.27
Other race, non-Hispanic	2.16 (0.25–18.72)	0.49	-	-
Hispanic, any race	Ref		Ref	
Marital status				
Single	6.07 (2.15–17.20)		12.23 (2.29–65.45)	0.003
Married	Ref		Ref	
Cancer stage				
Locally advanced	1.91 (1.00–3.67)	0.05	-	-
Localized	1.31 (0.48–3.59)	0.60	-	-
Recurrent	-	-	-	-
Advanced	-	-	-	-
Metastasis	Ref		-	
Risk factors of NMOU				
History of illicit drug use	6.30 (3.42–11.63)	<0.001	1.81 (0.66–4.96)	0.25
History of marijuana use	3.23 (1.78–5.86)	<0.001	1.98 (0.76–5.12)	0.16
History of tobacco use	5.45 (2.40–12.38)	<0.001	3.34 (1.12–9.99)	0.03
History of alcohol use	1.85 (1.00–3.44)	0.05	0.29 (0.10–0.79)	0.02
History of depression	1.42 (0.71–2.84)	0.32	-	-
History of bipolar disorder	3.01 (0.79–11.52)	0.11	-	-
History of schizophrenia	9.17 (1.80–46.73)	0.008	7.72 (0.94–63.17)	0.06
Family history of illicit drug use	3.89 (0.97–15.56)	0.06	-	-
Personal history of criminal activity	8.69 (4.52–16.71)	<0.001	0.86 (0.27–2.76)	0.80
Contact with persons involved in criminal activity	13.8 (6.70–28.41)	<0.001	9.86 (2.75–35.38)	<0.001
Inconsistent pain presentation	82.48 (10.07–675.43)	<0.001	-	-
Use for non-malignant pain	6.90 (2.50–19.06)	<0.001	7.45 (1.80–30.90)	0.006
Others **	27.64 (2.82–271.50)	0.004	-	-
MEDD at consultation	1.00 (0.99–1.00)	0.70	-	-
ESAS at consult only				
Pain	1.15 (1.03–1.28)	0.01	1.20 (1.04–1.38)	0.01
Tiredness	1.05 (0.95–1.16)	0.32	-	-
Nausea	1.07 (0.97–1.18)	0.15	-	-
Depression	1.11 (1.02–1.22)	0.02	-	-
Anxiety	1.11 (1.02–1.21)	0.02	-	-
Drowsiness	0.99 (0.90–1.10)	0.92	-	-
Appetite	1.12 (1.02–1.22)	0.02	-	-
Well-being	1.05 (0.95–1.15)	0.35	-	-
Shortness of breath	1.09 (1.00–1.20)	0.06	-	-
Family distress	1.11 (1.02–1.22)	0.02	-	-
Spiritual distress	1.07 (0.95–1.20)	0.30	-	-
Constipation	1.02 (0.94–1.12)	0.59	-	-
Sleep	1.05 (0.96–1.15)	0.25	-	-

** Others: homelessness, history of sexual abuse. Abbreviations: ESAS, Edmonton Symptom Assessment System; MEDD, morphine equivalent daily dose; NMOU, non-medical opioid use; PPA, Patient Prescriber Agreement.

## Data Availability

The data presented in the study are available on request from the corresponding author, SP. The data are not publicly available due to their containing information that could compromise the privacy of research subjects.

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
