# Peer review of "Adherence to Opioid Patient Prescriber Agreements at a Safety Net Hospital"

_cancers, 2023, doi:10.3390/cancers15112943_

Round 1

Reviewer 1 Report

The article has an appropriate research design, with well documented methods.

The sample is well distributed, with no bias.

Please evaluate and discuss the correlations between the answers given by patients to PPAs' questions and non-adherence. 

Author Response

The article has an appropriate research design, with well documented methods.

The sample is well distributed, with no bias.

REPLY: Thank you for your meticulous review.

Please evaluate and discuss the correlations between the answers given by patients toPPAs' questions and non-adherence.

REPLY: We apologize the confusion.  The purpose of this study was only to examine factors associated with non-adherence PPAs. “Among the patients who did not complete PPAs, we could not tell from the chart if it was because PPAs were not offered or patient refused to sign.  Thus, we could not evaluate the effectiveness of PPAs in this study.”  Further research is needed to examine this issue. 

Reviewer 2 Report

Line 26 : Patient prescriber agreements (PPAs, i.e., opioid contracts) is confusing, Please write differently for abbreviation and examples

P value should italic in complete manuscript.

Line 295:  "our resource-limited setting " , conclusions and limitations should be separate.

Better to add limitations and strengths of study. 

Author Response

Line 26 : Patient prescriber agreements (PPAs i.e., opioid contracts) is confusing, Please write differently for abbreviation and examples.

REPLY: Thank you for your comment. We have made the following changes.

“Patient prescriber agreements, also known as opioid contracts or opioid treatment agreement,   have been recommended as a strategy to mitigate non-medical opioid use (NMOU).”

P value should italic in complete manuscript.-  Thank you, we have done that.

Line 295: "our resource-limited setting " , conclusions and limitations should be separate.

REPLY: Based on your input, we have now revised the discussion to add more strengths and limitations as follows:

“Our results underscore that PPAs can be easily done in large sample size, even in resource limited settings, a substantial minority had NMOU behaviors and non-adherence to PPAs and we identified key patient risk factors associated with PPA non-adherence.  

Given our high-risk population, we believe universal PPAs should be promoted as one risk mitigation strategy to promote safe opioid use. Although the evidence that PPAs reduce adverse opioid-related events is limited, PPAs can be helpful as educational material, setting patient expectations for testing requirements and providing reassurance on the partnership, especially when many patients have an opioid phobia. PPAs can be used as a form of monitoring and documenting NMOU behavior. Furthermore, implementing a more universal protocol could help reduce clinicians’ implicit bias in providing opioid pain management.  

Our study has several strengths, including consecutive patient data collection, a large ethnically diverse sample size, and a unique focus on a cancer palliative care setting in a safety net hospital. However, our study had several limitations. There was a lack of generalizability, as the data was obtained from a single palliative care clinic in a single hospital system. Additionally, we could not determine if PPA reduces NMOU behavior or overdose.   The retrospective study design did not allow us to capture important predictors, such as patients’ attitudes and beliefs regarding PPAs.  Among the patients who did not complete PPAs, we could not tell from the chart if it was because PPAs were not offered or patient refused to sign.  Thus, we could not evaluate the effectiveness of PPAs in this study.  A prospective study may improve data collection, but having to obtain patient consent may result in selection bias. Future research, such as randomized controlled trials, is needed to assess the benefit of universal PPAs in reducing NMOU. In this era of the opioid epidemic, the management of patients on opioids is highly complex, and PPAs may potentially represent a low-cost and simple intervention among other risk-mitigation strategies to support safe opioid use. “

Reviewer 3 Report

The Authors present a good and interesting paper :" Adherence to opioid patient prescriber agreement at a safety net hospital".

The manuscript contains an appropriate Introduction as well as Material and Methods section. A good number of patients is reported (905) and a valid Statistical Analysis was performed.

Tables are too many and not easy to read.

Discussion: too long to me reporting too many previous studies not conclusive and creating confusion in  understanding by readers. I suggest to:

1. simplify the discussion shortening it and underlying what was really detected as important in their analysis

2. page 9 of 13 the last 2 paragraphs should be removed and inserted in the Conclusion. I am disappointed to read in a paper (so well done and reporting a large number of patients) that "further studies are needed " to better understand and assess the benefit of universal PPAs in reducing NMOU.

3. please clearly sintetize and underline the positive results of this research and what you suggest at the present as workable activity to be adopted to face this really important and relevant problem

Author Response

The Authors present a good and interesting paper :" Adherence to opioid patient prescriber agreement at a safety net hospital".

The manuscript contains an appropriate Introduction as well as Material and Methods section. A good number of patients is reported (905) and a valid Statistical Analysis was performed.

REPLY: Thank you for your meticulous review.

Tables are too many and not easy to read.

REPLY: Thank you for your comment.  We have reviewed the tables again and believe the data tables are essential to present our findings properly.  We thought of breaking Table 1 into two but will keep it for now for consistency.  We would be happy to be adjust if the editor feels this is necessary.

Discussion: too long to me reporting too many previous studies not conclusive and creating confusion in  understanding by readers. I suggest to:

  1. simplify the discussion shortening it and underlying what was really detected as important in their analysis

REPLY: Thank you.  We have now shortened the discussion to make it more concise.

  1. page 9 of 13 the last 2 paragraphs should be removed and inserted in the Conclusion. I am disappointed to read in a paper (so well done and reporting a large number of patients) that "further studies are needed " to better understand and assess the benefit of universal PPAs in reducing NMOU.

REPLY: Thank you.  We have now moved those 2 paragraphs to the conclusion.  We have also  stated in limitations (p.10, paragraph 3) that the outcomes,  are not examined and thus further research is needed:

“However, our study had several limitations. There was a lack of generalizability, as the data was obtained from a single palliative care clinic in a single hospital system. Additionally, we could not determine if PPA reduces NMOU behavior or overdose.   The retrospective study design did not allow us to capture important predictors, such as patients’ attitudes and beliefs regarding PPAs.  Among the patients who did not complete PPAs, we could not tell from the chart if it was because PPAs were not offered or patient refused to sign.  Thus, we could not evaluate the effectiveness of PPAs in this study.  A prospective study may improve data collection, but having to obtain patient consent may result in selection bias. Future research, such as randomized controlled trials, is needed to assess the benefit of universal PPAs in reducing NMOU. In this era of the opioid epidemic, the management of patients on opioids is highly complex, and PPAs may potentially represent a low-cost and simple intervention among other risk-mitigation strategies to support safe opioid use.” 

  1. please clearly synthetize and underline the positive results of this research and what you suggest at the present as workable activity to be adopted to face this really important and relevant problem

REPLY: Thank you.  We reported in our conclusion that PPAs can be done in a large sample size even in resource limited setting. Thus we believe that everyone should be able to adopt a form of PPA in their palliative practice.

“Given our high-risk population, we believe universal PPAs should be promoted as one risk mitigation strategy to promote safe opioid use. Although the evidence that PPAs reduce adverse opioid-related events is limited, PPAs can be helpful as educational material, setting patient expectations for testing requirements and providing reassurance on the partnership, especially when many patients have an opioid phobia. PPAs can be used as a form of monitoring and documenting NMOU behavior. Furthermore, implementing a more universal protocol could help reduce clinicians’ implicit bias in providing opioid pain management.  “

Round 2

Reviewer 3 Report

The Authors answered in an efficient way the requested modifications. I don’t have to ask other